# VLA-Cache: Efficient Vision-Language-Action Manipulation via Adaptive Token Caching

**Siyu Xu[1], Yunke Wang[1], Chenghao Xia[1], Dihao Zhu[1], Tao Huang[2], Chang Xu[1]***

[1]School of Computer Science, University of Sydney, Australia
[2]John Hopcropt Center for Computer Science, Shanghai Jiao Tong University, China
`{s.xu,yunke.wang,c.xu}@sydney.edu.au,{cxia0515,dzhu0968}@uni.sydney.edu.au,`
`t.huang@sjtu.edu.cn`

## Abstract

Vision-Language-Action (VLA) models have demonstrated strong multi-modal reasoning capabilities, enabling direct action generation from visual perception and language instructions in an end-to-end manner. However, their substantial computational cost poses a challenge for real-time robotic control, where rapid decision-making is essential. This paper introduces VLA-Cache, a training-free inference acceleration method that reduces computational overhead by adaptively caching and reusing static visual tokens across frames. Exploiting the temporal continuity in robotic manipulation, VLA-Cache identifies minimally changed tokens between adjacent frames and reuses their cached key-value representations, thereby circumventing redundant computations. Additionally, to maintain action precision, VLA-Cache selectively re-computes task-relevant tokens that are environmentally sensitive, ensuring the fidelity of critical visual information. To further optimize efficiency, we introduce a layer adaptive token reusing strategy that dynamically adjusts the reuse ratio based on attention concentration across decoder layers, prioritizing critical tokens for recomputation. Extensive experiments on two simulation platforms (LIBERO and SIMPLER) and a real-world robotic system demonstrate that VLA-Cache achieves up to **1.7× speedup** in CUDA latency and a **15% increase** in control frequency, with negligible loss on task success rate. The code and videos can be found at our project page: `https://vla-cache.github.io`.

## 1 Introduction

Learning a robust and generalizable policy for robotic manipulation through policy learning has long been a challenging problem [1], with traditional reinforcement learning approaches [2, 3] often suffering from poor robustness and limited generalization. Recently, the rapid advancement of foundational Vision-Language Models (VLMs) [4, 5] has demonstrated remarkable capabilities in multimodal understanding and generalization. Leveraging large-scale real-world robotic datasets [6, 7], pioneering works [8–11] have introduced Vision-Language-Action (VLA) models, which integrate vision and language modalities to directly generate robotic actions in an end-to-end manner. This emerging paradigm holds great promise for enhancing the adaptability and generalization of robotic control systems, but leaves a large computational demand.

To mitigate the extensive cost of VLA models, existing works often adopt generic acceleration techniques, such as model lightweighting [12], quantization [13], and early-exit [14]. While effective to some extent, these methods often require architectural modifications or retraining, and more importantly, they *lack task-specific design tailored to the intrinsic characteristics of VLA tasks*. As a

---

*Corresponding author.

39th Conference on Neural Information Processing Systems (NeurIPS 2025).

result, they struggle to achieve an optimal balance between inference speed and action accuracy. In this paper, we step into the nature of VLA robotic manipulation that is intrinsically different from the VLM models. Specifically, VLA tasks involve sequentially processing a stream of temporally adjacent visual observations, where the environment often exhibits high spatial redundancy across time. As shown in Figure 1, large portions of the visual scene, especially background regions, remain static and semantically irrelevant to action decisions, yet are processed repeatedly at each time step. These static tokens contribute significantly to computational overhead while providing limited utility for downstream control. This motivates our proposed token caching mechanism, which explicitly exploits temporal redundancy in visual inputs to reduce redundant computation without compromising decision quality.

To address the inefficiency introduced by repeatedly processing static visual information, we present **VLA-Cache**, a training-free inference acceleration method that exploits temporal continuity in robotic perception. Rather than recomputing all vision tokens at every timestep, VLA-Cache identifies tokens that exhibit minimal change between adjacent frames and reuses their cached key-value (KV) representations to bypass redundant computation. However, we observe that not all visually static tokens can be safely reused. Some tokens, such as those near the gripper or target object, may appear visually unchanged but remain semantically active and crucial for accurate action generation. Naively reusing all static tokens results in a significant performance drop as shown in Table 1. To mitigate this, VLA-Cache

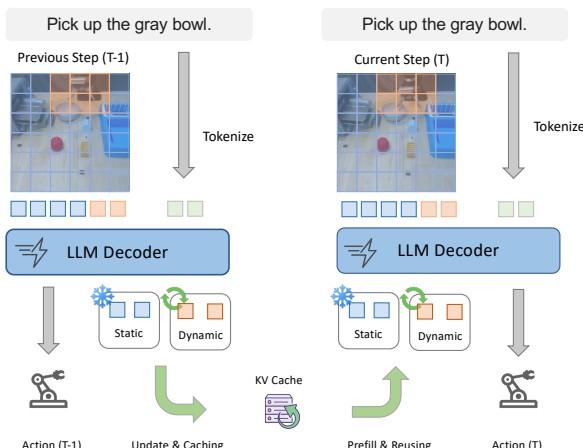

Figure 1: During the inference of the VLA model, static tokens of the input image remain largely consistent across steps. This consistency allows for caching the computations of these tokens from the previous step.

incorporates a lightweight filtering mechanism based on decoder attention scores to exclude task-relevant tokens from reuse, ensuring that semantically critical regions are always recomputed with up-to-date features. Moreover, we observe that attention patterns vary across decoder layers, with deeper layers exhibiting more concentrated focus. To further optimize reuse, VLA-Cache employs a layer-adaptive caching strategy that dynamically adjusts the reuse ratio per layer based on attention entropy, prioritizing precise updates in sensitive regions. These two mechanisms together enable substantial reduction in decoding overhead, especially in large-scale language decoders (*e.g.*, LLaMA [15], Gemma [16]), which typically dominate the compute cost in VLA systems.

The resulting method **VLA-Cache** offers a training-free and plug-and-play solution for accelerating VLA models without sacrificing action performance. We evaluate VLA-Cache on robotic manipulation tasks across two simulated environments (LIBERO [17] and SIMPLER [18]) and three state-of-the-art VLA models (OpenVLA [11], CogAct [19], and OpenVLA-OFT [20]). VLA-Cache consistently delivers over **1.7× acceleration** with only minor drops in task success rate. Furthermore, we demonstrate its real-world applicability by deploying it on a *Kinova Jaco2 robot arm*, achieving practical speedup under real-time control scenarios.

## 2    Related Work

**Vision-Language-Action Models.** Large-scale vision-language models (VLMs) have significantly advanced multimodal learning by integrating image understanding and language reasoning [5, 21]. Extending these capabilities, VLA models [8, 22] incorporate an action modality, enabling end-to-end visuomotor control. These models typically adopt large VLM backbones [15] and fine-tune them on robot data [6], with approaches varying from discretizing actions as language-like tokens [11, 23] to incorporating specialized diffusion policy heads [24]. Despite their effectiveness in tasks like object retrieval and assembly [25, 26], VLA models demand substantial computation, making real-time deployment challenging, particularly in resource-constrained environments.

**Acceleration for Vision-Language Models.** Inference acceleration has been extensively explored in vision-language models (VLMs) through quantization [27], pruning [28], and token-level techniques such as FastV [29], SparseVLM [30], ToMe [31], PuMer [32], and MADTP [33]. These intra-frame strategies reduce redundancy within a **single image** but disregard the temporal and spatial structure essential for robotic tasks under closed-loop control. In the VLA domain, efficiency has been addressed through architectural modifications (*e.g.*, RoboMamba [34], TinyVLA [12]), quantization-aware training (QAIL [13]), and dynamic depth control (DeeR-VLA [14]). While effective, these methods require re-training and lack generalizability. Recent high-frequency frameworks such as $\pi_0$-FAST [35], HiRT [36], and OpenVLA-OFT [20] achieve higher control frequency via action chunking or asynchronous decoding. However, they continue to suffer from the **language model decoding** bottleneck, which dominates inference time. VLA-Cache addresses this gap by introducing a **cross-frame** token reuse strategy that accelerates inference without modifying the model or requiring additional training. Furthermore, it complements existing high-frequency VLA architectures by directly accelerating the language decoder bottleneck, offering a lightweight, plug-and-play solution for real-time robotic inference.

## 3 Methodology

In robotic action prediction, most visual tokens remain static across frames except for key regions like the manipulator or target object. While this temporal redundancy enables token reuse, reusing all static tokens can harm accuracy when task-relevant regions subtly change. To address this, we propose a method that identifies visually static tokens and filters out semantically important ones based on attention scores from the VLA decoder. By avoiding redundant computation of unchanged static tokens between adjacent frames, our approach directly alleviates the computation bottleneck of language decoder in VLA models while preserving the accuracy of action prediction.

### 3.1 KV Cache for VLA Token Reusing

Key-Value (KV) caching is a widely adopted technique in large-scale autoregressive models to reduce both computation and memory footprint during decoding. Initially proposed in the Transformer architecture [37], KV caching enables the model to reuse previously computed key ($\mathbf{K}$) and value ($\mathbf{V}$) vectors for each token, thereby avoiding redundant computation across decoding steps. Concretely, given a sequence of input tokens $\mathbf{X}$, the self-attention mechanism computes:

$$\mathbf{Q} = \mathbf{X}W_Q, \quad \mathbf{K} = \mathbf{X}W_K, \quad \mathbf{V} = \mathbf{X}W_V, \tag{1}$$

$$\text{Attn}(\mathbf{Q}, \mathbf{K}, \mathbf{V}) = \text{Softmax}\left(\frac{\mathbf{Q}\mathbf{K}^\top}{\sqrt{d}}\right)\mathbf{V}. \tag{2}$$

During decoding, each new token's $\mathbf{k}_{\text{new}}$ and $\mathbf{v}_{\text{new}}$ are appended to the $i$-th cache:

$$\mathbf{K}_i = \text{Concat}(\mathbf{K}_{i-1}, \mathbf{k}_{\text{new}}), \quad \mathbf{V}_i = \text{Concat}(\mathbf{V}_{i-1}, \mathbf{v}_{\text{new}}). \tag{3}$$

While KV caching is effective for language decoding within a single query in vision-language models, this technique does not address redundancy in the visual stream, especially in Vision-Language-Action (VLA) models. In robotic manipulation, consecutive visual inputs often share large overlapping content, yet VLA models typically discard visual encodings after each step and recompute them from scratch. This is both wasteful and suboptimal for real-time control.

This inefficiency motivates a key question: *can we selectively reuse static visual tokens across time with temporal KV Caching in VLA models?* This idea forms the basis of our proposed **VLA-Cache**. In the following sections, we introduce its core mechanisms: static token selection, task-relevance filtering, and layer-adaptive reuse to accelerate VLA inference while preserving action accuracy.

### 3.2 Temporal Redundancy in Robotic Perception

In closed-loop robotic manipulation, consecutive visual frames often share large portions of static content. As illustrated in Figure 2, background regions and stationary objects typically exhibit negligible changes between adjacent frames. However, most existing Vision-Language-Action (VLA)

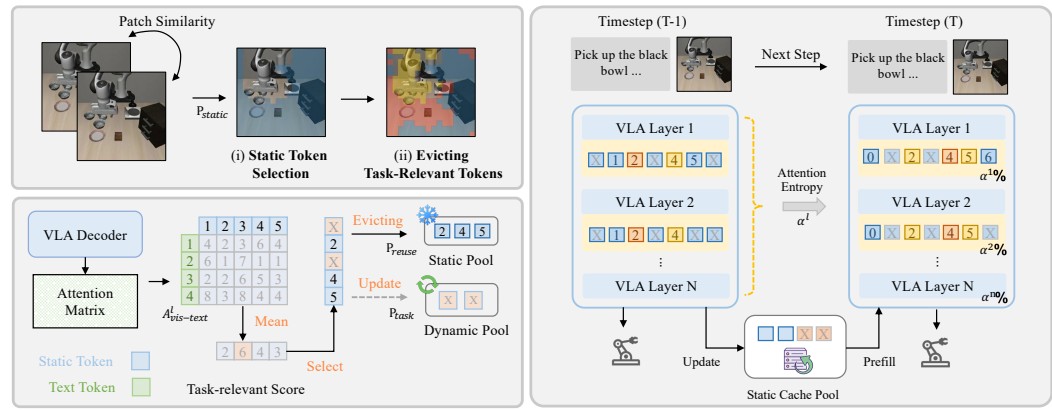

| (a) Dynamic Token Selection | (b) Adaptive Token Caching |

Figure 2: **VLA-Cache** accelerates the VLA's language decoding process across timesteps via the following two steps: (a) *Dynamic Token Selection* reuses static tokens across frames while preserving task-relevant ones; (b) *Adaptive Token Caching* dynamically adjusts reuse ratios per decoder layer based on attention patterns.

models discard visual representations after each timestep and recompute all visual tokens from scratch, resulting in substantial redundancy and increased inference latency.

To address this inefficiency, we propose selectively reusing visual tokens that remain static across timesteps. Specifically, we identify and cache the representations of image regions with minimal visual change, allowing their Key-Value (KV) representations to be reused in the next frame. This strategy significantly reduces redundant computation in the VLA visual stream while preserving model performance.

**Static Token Selection.** Given an image $I \in \mathbb{R}^{H \times W \times 3}$, we divide it into $N \times N$ non-overlapping patches of size $p \times p$, yielding a set of raw pixel patches $\mathcal{P}_t = \{\mathbf{P}_t^{i,j}\}$. For each patch $\mathbf{P}_t^{i,j}$ in the current frame and its corresponding patch $\mathbf{P}_{t-1}^{i,j}$ in the previous frame, we compute cosine similarity:

$$\mathrm{Sim}\big(\mathbf{P}_t^{i,j}, \mathbf{P}_{t-1}^{i,j}\big) = \frac{\mathbf{P}_t^{i,j} \cdot \mathbf{P}_{t-1}^{i,j}}{\|\mathbf{P}_t^{i,j}\|_2 \cdot \|\mathbf{P}_{t-1}^{i,j}\|_2}. \tag{4}$$

A patch is considered visually static if its similarity exceeds a threshold $\tau$. We further apply a Top-$k$ filter to retain the most stable tokens:

$$\mathcal{P}_{\mathrm{static}} = \text{Top-}k\Big(\big\{\mathbf{P}_t^{i,j} \mid \mathrm{Sim}(\mathbf{P}_t^{i,j}, \mathbf{P}_{t-1}^{i,j}) \geq \tau\big\}\Big). \tag{5}$$

This simple yet effective approach accurately selects truly static tokens across consecutive frames, significantly reducing redundant computations and accelerating inference without compromising overall performance.

### 3.3 Retaining Task-Relevant Information

While visual similarity offers a practical signal for reusing static regions, not all visually static tokens are safe to reuse. In robotic control tasks, certain regions, such as the gripper or target object, though visually unchanged, are semantically dynamic and critical for precise action generation. Directly reusing all visually static tokens, without considering their semantic role, can lead to serious performance degradation. As shown in Table 1, naive static token reuse lowers the success rate from the OpenVLA baseline of **84.4%** to just **74.2%**.

Table 1: Comparison of VLA-Cache's core token selection strategies on OpenVLA using the LIBERO Spatial benchmark.

| Method | SR (%) ↑ | Latency (ms) ↓ |
|---|---|---|
| OpenVLA | 84.4 | 51.56 |
| + Static Token | 74.2 | 31.03 |
| + Evict Task-Relevant | 82.6 | 31.03 |
| + Layer Adaptive | 83.8 | 32.22 |

This degradation occurs because task-relevant tokens, though minimally changed at the pixel level, exhibit greater feature sensitivity to subtle environmental changes. Unlike vision-language models

| **Algorithm 1** Dynamic Token Selection | **Algorithm 2** Adaptive Token Caching |
|---|---|
| 1: **Input:** Frames $\{I_{t-1}, I_t\}$, thresholds $\tau$, $\tau_{\text{task}}$, hyperparameter $k$ | 1: **Input:** Token indices $\mathcal{P}_{\text{final}}$, previous KV cache $\{\mathbf{K}_{t-1}^l, \mathbf{V}_{t-1}^l\}$, representations $\mathbf{H}_t^l$ |
| 2: **Output:** Reusable token indices $\mathcal{P}_{\text{final}}$ | 2: **Output:** Updated KV cache $\{\mathbf{K}_t^l, \mathbf{V}_t^l\}$ |
| 3:        ▷ *Static Token Selection* | 3: Compute entropy $R_{\text{cum}}^l$, reuse ratio $\alpha^l$, subset $\mathcal{P}_{\text{reuse}} \subseteq \mathcal{P}_{\text{final}}$ |
| 4: Patchify both frames $I_{t-1}, I_t$ and compute $\text{Sim}(\mathbf{P}_t, \mathbf{P}_{t-1})$ among related patches | 4: **for** each layer $l$ and token $i$ **do** |
| 5: Select $\mathcal{P}_{\text{static}}$ where similarity $\geq \tau$ | 5:     **if** $i \in \mathcal{P}_{\text{reuse}}$ **then** |
| 6: Apply top-$k$ filtering to refine static token selection | 6:        Reuse cached values: $\mathbf{K}_t^l(i) = \mathbf{K}_{t-1}^l(i), \mathbf{V}_t^l(i) = \mathbf{V}_{t-1}^l(i)$ |
| 7:        ▷ *Evict Task-Relevant Tokens* | 7:     **else** |
| 8: Compute text-to-vision attention scores $\mathbf{S}_{\text{task-relevance}}$ | 8:        Recompute: $\mathbf{K}_t^l(i) = W_K^l \mathbf{H}_t^l(i)$, $\mathbf{V}_t^l(i) = W_V^l \mathbf{H}_t^l(i)$ |
| 9: Select $\mathcal{P}_{\text{task-relevant}}$ where attention $\geq \tau_{\text{task}}$ | 9:     **end if** |
| 10: Compute reusable tokens: $\mathcal{P}_{\text{final}} = \mathcal{P}_{\text{static}} \setminus \mathcal{P}_{\text{task-relevant}}$ | 10: **end for** |
| 11: **return** $\mathcal{P}_{\text{final}}$ | 11: **return** $\{\mathbf{K}_t^l, \mathbf{V}_t^l\}$ |

(VLMs), VLA models must track object states and interactions over time, making them more dependent on accurate visual encoding. Therefore, to ensure alignment with the latest environment state, task-relevant regions must be recomputed each step.

**Evicting Task-Relevant Tokens.** To avoid reusing semantically critical but visually static tokens, we propose a lightweight filtering mechanism using cross-attention scores from the language decoder. For each decoder layer $l$, we extract the text-to-vision attention matrix $\mathbf{A}_{\text{vis-text}}^l$ from the full attention tensor $\mathbf{A}^l \in \mathbb{R}^{N_{\text{heads}} \times N_{\text{tokens}} \times N_{\text{tokens}}}$ as:

$$\mathbf{A}_{\text{vis-text}}^l = \mathbf{A}^l[:, v_{\text{start}} : v_{\text{end}}, t_{\text{start}} : t_{\text{end}}], \tag{6}$$

where $v_{\text{start}}, v_{\text{end}}$ and $t_{\text{start}}, t_{\text{end}}$ are the indices of the vision and text tokens, respectively. To aggregate the attention scores across multiple heads, we compute the mean attention for each vision token as $\mathbf{A}_{\text{avg}}^l = \text{Mean}_{\text{heads}}(\mathbf{A}_{\text{vis-text}}^l)$. For task relevance across multiple layers $\mathcal{L}$, the final task relevance scores are obtained by averaging the scores across the selected layers as $\mathbf{S}_{\text{task-relevance}} = \text{Mean}_{l \in \mathcal{L}}(\mathbf{A}_{\text{avg}}^l)$. Using these scores, we rank the vision tokens based on their task relevance and apply a threshold $\tau_{\text{task}}$ to select the most task-relevant tokens:

$$\mathcal{P}_{\text{task-relevant}} = \{\mathbf{P}_t^{i,j} \mid \mathbf{S}_{\text{task-relevance}}[i, j] \geq \tau_{\text{task}}\}. \tag{7}$$

Finally, we combine the set of static tokens $\mathcal{P}_{\text{static}}$ selected in the first step with the task-relevant tokens. Tokens that are both static and highly task-relevant are removed from the reusable token set to ensure they are recomputed in the current step:

$$\mathcal{P}_{\text{reuse}} = \mathcal{P}_{\text{static}} \setminus \mathcal{P}_{\text{task-relevant}}. \tag{8}$$

By filtering out semantically significant tokens from the static reuse set, our method restores the degraded success rate from **74.2%** to **82.6%**, while maintaining the computational gains in FLOPs and latency. This balance between task fidelity and efficiency illustrates the value of cross-modal attention as a lightweight signal for safe token reuse in VLA models.

## 3.4 Layer Adaptive Token Reusing

While static token selection and task-relevance filtering eliminate a large portion of redundant computation, we observe that attention distributions within the VLA decoder vary significantly across different layers. This finding is consistent with observations reported by prior work *FastV* [29], indicating that both VLA and VLM decoders exhibit similar patterns of attention flow: early layers display dispersed attention, followed by fluctuations in intermediate layers, and eventually a partial rebound near the final layers.

To account for these differences, we propose a *layer-adaptive strategy* that adjusts the fraction of reused tokens based on each layer's attention concentration. Specifically, we quantify the attention

distribution at layer $l$ via an *entropy* measure, following the same mean-attention computation described in Eq. 6. Let $\mathcal{E}^l$ denote the resulting entropy. We then define an *entropy ratio* $R^l = (\mathcal{E}^{l-1} - \mathcal{E}^l)/\mathcal{E}^{l-1}$, which captures how much more concentrated the attention is in layer $l$ compared to layer $l - 1$.

A positive $R^l$ indicates that the attention distribution at layer $l$ is more focused than that of layer $l - 1$. We accumulate these ratios across layers to obtain a cumulative score, which in turn determines the proportion $\alpha^l$ of static tokens (from $\mathcal{P}_{\text{final}}$) that are reused at layer $l$. Formally,

$$\alpha^l \;=\; \min\Big(k\sum_{j=1}^{l} R^j,\; 1\Big), \tag{9}$$

where $k$ is a hyperparameter that governs the impact of attention concentration. Layers with larger cumulative entropy reduction are allowed to reuse a higher fraction of tokens, reflecting the insight that as attention becomes more focused, fewer tokens are likely to require recomputation.

In practice, this layer-adaptive mechanism dynamically adjusts token reuse based on the evolving attention patterns in the VLA decoder, effectively balancing computational efficiency with task accuracy. By selectively retaining only the most relevant tokens at each layer, our method significantly reduces redundant computations while maintaining reliable action prediction.

## 4 Implementations

### 4.1 Cross-Frame Visual Token Caching

During inference, VLA-Cache accelerates robotic action prediction by reusing previously computed key-value (KV) representations of visual tokens across time. Instead of recomputing all visual tokens in each time step, the model identifies a subset of static tokens, those that exhibit minimal change across frames, and reuses their cached representations from the previous time step. In contrast, dynamic tokens, which undergo significant visual change or are task-relevant, are freshly computed to maintain accurate action generation.

At each timestep $t$, given the visual token sequence $\mathbf{H}_t$, the model identifies a subset $\mathcal{P}_{\text{reuse}}$ of tokens that remain unchanged from the previous frame and reuses their cached representations:

$$\mathbf{K}_t(i) = \begin{cases} \mathbf{K}_{t-1}(i), & i \in \mathcal{P}_{\text{reuse}} \\ W_K\mathbf{H}_t(i), & \text{otherwise} \end{cases}, \quad \mathbf{V}_t(i) = \begin{cases} \mathbf{V}_{t-1}(i), & i \in \mathcal{P}_{\text{reuse}} \\ W_V\mathbf{H}_t(i), & \text{otherwise} \end{cases}. \tag{10}$$

This design avoids redundant computation for static tokens while ensuring dynamic or task-relevant inputs are freshly computed. Most existing approaches reduce computation by pruning or merging tokens within a single frame [29–33]. In contrast, VLA-Cache exploits temporal redundancy by caching and reusing visual tokens across frames, making it better aligned with the closed-loop nature of robotic control. Notably, it is compatible with high-frequency architectures and directly alleviates the *decoding bottleneck*. Since it requires no model modification or retraining, VLA-Cache serves as a *plug-and-play* optimization for efficient robotic inference. An overview of the inference procedure is shown in Algorithm 1 and Algorithm 2, detailing token selection and adaptive caching. Additional implementation details are provided in Appendix D.

### 4.2 Theoretical Analysis of Computational Complexity

**Overhead of Token Selection.** The cost of static token identification is approximately $\mathcal{O}(H^2)$ due to patch similarity checks, while task-relevance filtering introduces a cross-modal attention aggregation cost of $\mathcal{O}(L_t L_v D)$. The entropy-based layer-adaptive strategy incurs an additional $\mathcal{O}(L^2 D)$ complexity, which remains significantly lower than the baseline per-layer cost.

**Computational Cost Reduction.** In standard VLA inference, each Transformer layer processes $L$ tokens, with a total FLOP cost per layer:

$$\text{FLOPs} \approx 4LD^2 + 2L^2D + 2LDM. \tag{11}$$

VLA-Cache reduces effective token count per layer to $L_r = \alpha \times \mathcal{P}_{\text{final}}$, leading to theoretical savings:

$$\Delta\text{FLOPs}_{\text{layer}} \approx 4L_r D^2 + 2L_r^2 D + 2L_r DM. \tag{12}$$

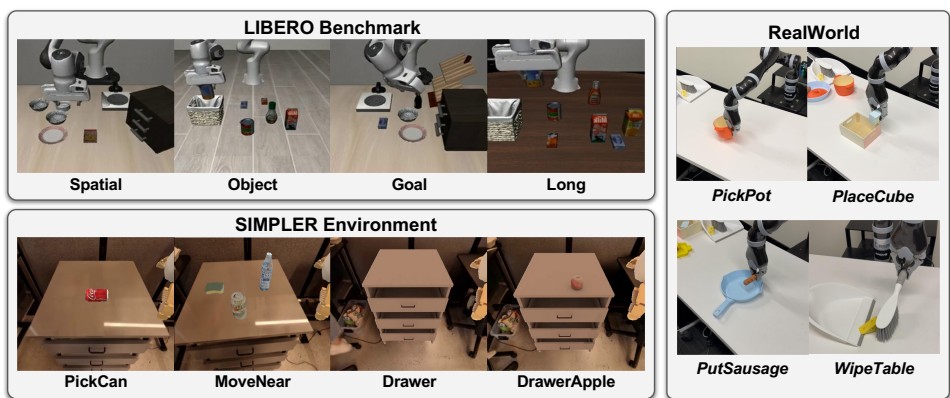

Figure 3: Tasks on LIBERO Benchmark, the SIMPLER Environment and Real World.

**Total Complexity Reduction.** Bringing all components together, the theoretical overall FLOP reduction per layer is:

$$\Delta\text{FLOPs}_{\text{total}} \approx \left(4L_r D^2 + 2L_r^2 D + 2L_r DM\right) - \left(H^2 + L_t L_v D + L^2 D\right). \tag{13}$$

Please refer to Appendix C for detailed derivations, including static token selection costs, attention filtering complexity, and layer-adaptive entropy calculations.

## 5 Experiment

To validate the effectiveness of VLA-Cache, we evaluate our method in both simulation and real-world settings. In simulation, we evaluate VLA-Cache on three open-source VLA models: OpenVLA [11], OpenVLA-OFT [20] and CogAct [19], using the LIBERO benchmark [17] and SIMPLER environment [18], respectively. All experiments are conducted on an NVIDIA RTX 4090 GPU.

### 5.1 Experiment Setup

**Compared Methods.** We leverage the architectural similarity between VLA and VLM models, which allows direct application of existing VLM acceleration methods to VLA inference. Specifically, we adopt two state-of-the-art token-level acceleration techniques SparseVLM [30] and FastV [29] on OpenVLA as compared methods in the LIBERO benchmark.

**Evaluation Metrics.** We evaluate VLA-Cache using four metrics: success rate, control frequency, FLOPs, and CUDA latency. Success rate and control frequency respectively assess task performance and the responsiveness of action prediction in closed-loop control. FLOPs measure theoretical computation, while CUDA latency captures actual GPU runtime. These two efficiency metrics are widely adopted in VLM/VLA acceleration methods.

### 5.2 Evaluation Benchmark

**LIBERO.** The LIBERO Benchmark [17] covers four task suites: Spatial, Object, Goal, and Long, each testing a different aspect of manipulation generalization. We follow the standard setup from OpenVLA [11] and OpenVLA-OFT [20], using official weights and machines for consistency. Each suite includes ten subtasks evaluated over multiple episodes.

**SIMPLER.** The SIMPLER simulator [18] offers two settings, *Visual Matching* and *Variant Aggregation*, designed to bridge simulation-to-reality gaps. Following CogAct's setup [19], we evaluate both settings on a Google robot arm across four manipulation tasks. CogAct, which integrates a Diffusion Policy for continuous control, serves as our baseline. These evaluations demonstrate the generality of VLA-Cache under different action heads and simulation variations.

**Real Robot Evaluation.** We deploy VLA-Cache on a Kinova Jaco2 manipulator equipped with a front-facing camera. The robot is evaluated on four tasks: *PickPot*, *PlaceCube*, *PutSausage*, and

Table 2: Comparison of different VLA acceleration methods on the LIBERO benchmark.

| Method | Success Rate ↑ | | | | | FLOPs (T)↓ | Latency (ms)↓ | Control Freq. (Hz)↑ |
|---|---|---|---|---|---|---|---|---|
| | Spatial | Object | Goal | Long | Average | | | |
| OpenVLA | 84.4% | 86.6% | 75.6% | 53.2% | 75.0% | 1.864 | 51.91 | 4.23 |
| + SparseVLM | 79.8% | 67.0% | 72.6% | 39.4% | 64.7% | 1.407 | 83.39 | 3.72 |
| + FastV | 83.4% | 84.0% | 74.2% | 51.6% | 73.3% | 1.864 | 53.28 | 4.19 |
| **+ VLA-Cache** | **83.8%** | **85.8%** | **76.4%** | **52.8%** | **74.7%** | **1.355** | **31.83** | **4.59** |
| OpenVLA-OFT | 97.8% | 97.6% | 97.6% | 94.2% | 96.8% | 4.013 | 79.05 | 65.10 |
| **+ VLA-Cache** | **98.3%** | 97.5% | **98.3%** | **95.4%** | **97.4%** | **3.097** | **62.59** | **78.98** |

Table 3: Comparison of VLA-Cache within the CogACT model in the SIMPLER environment.

| SIMPLER | Method | Success Rate ↑ | | | | | FLOPs (T)↓ | Latency (ms) ↓ | Control Freq. (Hz) ↑ |
|---|---|---|---|---|---|---|---|---|---|
| | | PickCan | MoveNear | Drawer | DrawerApple | Average | | | |
| Matching | CogACT | 91.3% | **85.0%** | **71.8%** | 50.9% | **74.8%** | 1.847 | 54.29 | 12.42 |
| | **+ VLA-Cache** | **92.0%** | 83.3% | 70.5% | **51.6%** | 74.4% | **1.496** | **39.63** | **14.66** |
| Aggregation | CogACT | 89.6% | 80.8% | 28.3% | **46.6%** | 61.3% | 1.807 | 53.54 | 12.36 |
| | **+ VLA-Cache** | **91.7%** | 79.3% | **32.5%** | 45.8% | **62.3%** | **1.493** | **39.11** | **14.48** |

*WipeTable*, with the last task including diverse distractor objects to test robustness. Demonstrations are collected via teleoperation at 10 Hz using an Xbox controller, resulting in 150-200 trajectories per task. We fine-tune OpenVLA with LoRA [38] and evaluate on the same tasks, using the LIBERO tuning setup for consistency. More details about real-world experiments are available in the AppendixE.4.

## 5.3 Results on Simulation Environment

**Main Results on LIBERO.** Table 2 summarizes results across the four LIBERO task suites. VLA-Cache reduces FLOPs by **27.31%** and improves latency by **1.63×** over standard OpenVLA, with only a 0.3% drop in success rate. It performs robustly across tasks and exceeds the baseline on goal-oriented manipulation. When applied to OpenVLA-OFT, a faster variant with action chunking, VLA-Cache further boosts control frequency by nearly **14 Hz**, showing strong compatibility with high-frequency architectures and delivering additive gains even on optimized VLA models. In contrast, FastV and SparseVLM fail to improve inference speed and often degrade task performance. Their token pruning and merging strategies operate within a *single frame* and disrupt **spatial fidelity**, which is critical for precise manipulation. Moreover, these methods target long output sequences, whereas VLA models generate short action outputs (*e.g.*, 7 tokens), rendering the speedups marginal.

As illustrated in Figure 4, VLA-Cache effectively reduces visual computation redundancy during robotic manipulation by precisely identifying static tokens and filtering out task-relevant regions in real time. Notably, OpenVLA-OFT takes both *fixed third-person* and *dynamic wrist-camera* views as inputs. Its strong performance demonstrates that VLA-Cache not only enhances control frequency but also maintains robustness under **dynamic viewpoints** and camera shifts.

**Ablation on Token Reusing/Pruning Rate.** Table 4 presents an ablation study varying the number of reused/pruned tokens. For all methods, ag-

Table 4: Ablation on token pruning/reuse in LIBERO-Spatial using OpenVLA (256 vision tokens). Best values are in **bold**.

| #Tokens | Methods | SR % ↑ | FLOPs ↓ | Latency (ms) ↓ |
|---|---|---|---|---|
| 0 | Baseline | 84.4 | 1.888 | 52.37 |
| 50 | SparseVLM | 79.8 | **1.358** | 88.08 |
| | FastV | 84.6 | 1.888 | 53.10 |
| | Ours | **85.4** | 1.611 | **33.43** |
| 100 | SparseVLM | 74.6 | **1.097** | 61.01 |
| | FastV | 83.4 | 1.888 | 45.72 |
| | Ours | **83.8** | 1.295 | **31.29** |
| 200 | SparseVLM | 44.4 | **0.735** | 57.42 |
| | FastV | 72.8 | 1.888 | 45.19 |
| | Ours | **68.3** | 0.823 | **30.29** |

gressive token reduction harms success rate, underscoring the need to preserve informative content. VLA-Cache maintains stable performance at moderate reuse rates (*i.e.*, 100 tokens), while FastV and SparseVLM suffer larger drops due to loss of critical visual details. By directly updating KV entries, VLA-Cache remains more efficient and robust under different token reuse configurations.

**Token Selection Strategies.** As shown in Table 1, directly reusing all static tokens leads to a notable drop in success rate 74.2%, indicating that visual similarity alone is insufficient for reliable reuse in robotic control. By filtering out task-relevant tokens based on decoder attention, our method recovers

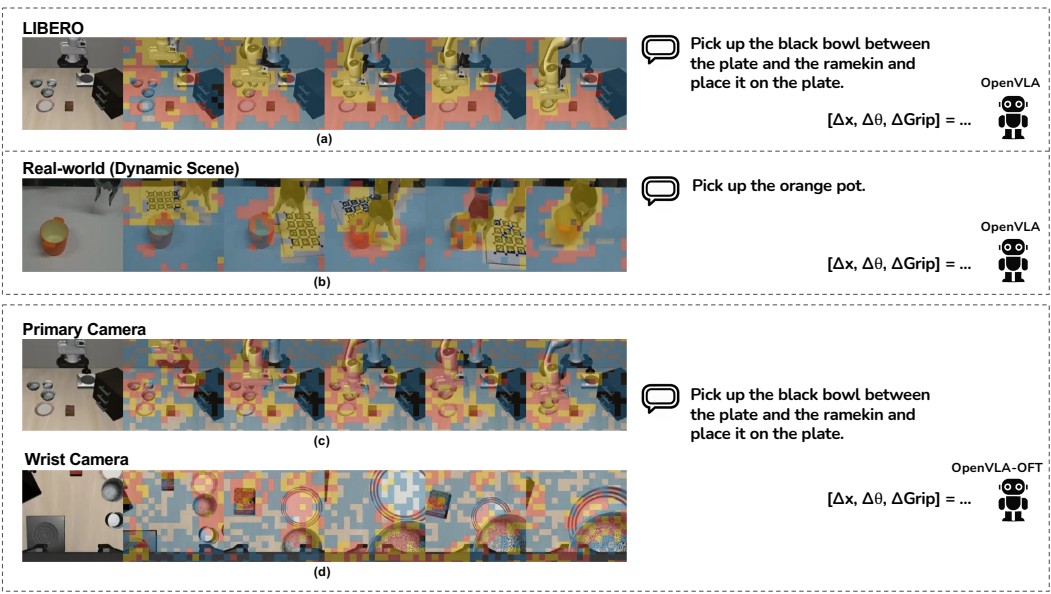

Figure 4: Visualization of VLA-Cache token reuse across settings. **(a)** LIBERO simulation with OpenVLA. **(b)** Real-world task under dynamic background. **(c)** and **(d)** Main and wrist camera views from OpenVLA-OFT. Blue: static tokens, Yellow: task-relevant, Red: overlapping. VLA-Cache reduces redundant computation and preserves accuracy under varying conditions.

performance to 82.6%. Introducing a layer-adaptive strategy further improves accuracy to 83.8%, with minimal increase in CUDA latency.

**Main Results on SIMPLER** Table 3 indicates that VLA-Cache exhibits success rates comparable to the CogACT baseline in the SIMPLER environment while substantially reducing computational overhead.

The efficiency gains are evident in the FLOPs and inference time measurements. VLA-Cache achieves roughly 20% fewer FLOPs than the baseline, coupled with a $1.37\times$ reduction in inference latency. Notably, these results highlight the portability of VLA-Cache across different action heads, establishing it as a general acceleration strategy for VLA.

Table 5: Comparison of success rate on real robot tasks.

| Method | Success Rate ↑ | | | | | FLOPs (T)↓ | Latency (ms) ↓ | Control Freq. (Hz) ↑ |
|---|---|---|---|---|---|---|---|---|
| | PickPot | PlaceCube | PutSausage | WipeTable | Average | | | |
| OpenVLA | **95.0%** | 83.3% | 80.0% | 70.0% | 82.1% | 1.814 | 64.16 | 4.02 |
| + VLA-Cache | 90.0% | **90.0%** | **85.0%** | **73.3%** | **84.6%** | **1.303** | **51.85** | **4.21** |

## 5.4 Results on Real Robot

Table 5 illustrates the performance of VLA-Cache in real-world robotic tasks. Among the four tasks, *PickPot* shows a slightly lower success rate than the baseline, whereas VLA-Cache exceeds the baseline on other three tasks. The method also achieves considerable reductions in FLOPs and inference time. Overall, VLA-Cache improves the average success rate by 2.4%, likely due to reduced interference from redundant visual tokens and enhanced decision robustness. With improved robustness as a foundation, VLA-Cache's ability to prune or reuse redundant tokens may further enhance the model's resilience, thus yielding higher success rates.

**Performance under Dynamic Background.** To assess robustness, we introduced background motion (e.g., human hands and moving objects) in the *PickPot* task. As shown in Table 7, success rate of baseline dropped from 95% to 80% under noise. With VLA-Cache, the same success rate was maintained while reducing FLOPs by 42% and latency by 35%. These results highlight VLA-Cache's ability to filter out transient or irrelevant tokens, preserving both efficiency and stability in dynamic real-world settings. Figure 4 visualizes this effect, showing robust token selection despite background disturbances.

## 6  Conclusion

In this paper, we introduce VLA-Cache, a training-free method for VLA that selectively reuses static tokens while filtering out task-relevant ones, reducing redundant computation without sacrificing accuracy. Additionally, our layer-adaptive token reuse strategy improves model success rates by adjusting token reuse based on attention concentration. Extensive experiments on three VLA models, OpenVLA, CogAct and OpenVLA-OFT, across two simulation environments, LIBERO and SIMPLER, demonstrate that VLA-Cache achieves a $1.7\times$ speedup while maintaining performance. Furthermore, we demonstrate its real-world applicability by deploying it on a Kinova Jaco2 robot arm, and VLA-Cache achieves practical speedup under real-time control scenarios.

## Acknowledgments

This work was supported in part by the Australian Research Council under Projects DP240101848 and FT230100549. We would also like to thank the anonymous reviewers and area chair for their constructive feedback, which helped improve the clarity and quality of this paper.

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

## A    Limitations

In this section, we discuss the potential limitations of the proposed VLA-Cache: (i) In dynamic environments with substantial background or object motion, the number of non-reusable tokens increases, reducing acceleration gains. As visualized in Figure 4, dynamic regions (highlighted in yellow) require full recomputation. (ii) Our experiments focus on three state-of-the-art open-source VLA architectures (OpenVLA [11], CogAct [19], OpenVLA-OFT [20]) based on LLaMA2 [15] decoders. The applicability of VLA-Cache to emerging VLA systems with different backbones (e.g., Gemma2 [16] in $\pi_0$ [24]) or more complex VLA systems remains an open direction for future work.

## B    Impact Statement

While VLA-Cache demonstrates robust performance even under dynamic conditions, we emphasize the importance of careful deployment and continuous monitoring to ensure safe, interpretable, and reliable behavior in real-world robotic systems.

## C    Complexity Analysis Details

**Static Token Selection.** We compute patch-wise similarity for visual tokens:

$$\text{FLOPs}_{\text{static-sim}} = N_{\text{patch}}^2 D_{\text{patch}} \approx H^2. \tag{14}$$

Since $N_{\text{patch}} = H/p$, this cost remains small relative to Transformer computations.

**Task-Relevance Filtering.** The attention-based filtering step computes cross-modal importance scores:

$$\text{FLOPs}_{\text{task-filter}} \approx L_t L_v D. \tag{15}$$

A sorting operation of $\mathcal{O}(L \log L)$ follows for threshold selection.

**Layer-Adaptive Entropy Computation.** The entropy-based reuse strategy involves:

$$\text{FLOPs}_{\text{entropy}} \approx L^2 D. \tag{16}$$

The per-layer reuse ratio is then computed as:

$$\alpha^l = \min\left(k \sum_{j=1}^{l} R^j, 1\right). \tag{17}$$

These overheads remain modest compared to the full forward pass cost, enabling efficient token reuse.

**Final FLOP Reduction.** The total FLOP savings across all layers follow:

$$\Delta\text{FLOPs}_{\text{total}} = \sum_{l=1}^{\Omega} \Delta\text{FLOPs}_{\text{layer}}. \tag{18}$$

This confirms that dynamic token reuse significantly reduces computation without sacrificing model performance.

## D    Inference Detail of VLA-Cache

**Inference Procedure.**    VLA-Cache accelerates robotic action prediction by reusing static visual tokens across frames during inference. At each timestep $t$, current visual tokens $\mathbf{H}_t$ are compared with those from the previous frame to identify unchanged regions. Tokens that are both visually static and task-irrelevant are reused via cached key-value (KV) entries, while task-relevant or dynamic tokens are recomputed. This selective reuse substantially reduces visual processing cost without altering model architecture or training.

**Token Reuse Mechanism.**   Our implementation modifies the VLA decoder's forward pass as follows:

- **Position and Attention Masking.** We maintain a `cache_position` array to mark tokens requiring recomputation. Static tokens retain their previous position encodings, allowing attention masks to be pruned to match the reduced token set.

- **Rotary Embedding.** For recomputed tokens, rotary embeddings are applied to introduce positional information. Tokens that are skipped retain their previous encoded states.

- **Dynamic Cache Updates.** Newly computed tokens update their respective $\{\mathbf{K}_t^l, \mathbf{V}_t^l\}$ entries, while reused tokens inherit values from the previous frame's cache $\{\mathbf{K}_{t-1}^l, \mathbf{V}_{t-1}^l\}$. Due to the permutation invariance of Transformers, this partial update yields valid attention results.

This strategy is fully compatible with standard KV caching in autoregressive decoding. The largest computational gain occurs when generating the first action token at each timestep; subsequent tokens are decoded autoregressively without additional cost.

**Experimental Settings.**   All experiments are conducted using OpenVLA with 256 visual tokens. Unless specified otherwise, we use a static token similarity threshold $\tau = 0.996$, top-$k = 100$ for retained static tokens, and a task-relevance threshold $\tau_{\text{task}} = 0.5$. These parameters are applied consistently across all simulated and real-world settings, including SIMPLER with CogAct. For real-world Jaco2 experiments, we slightly reduce the similarity threshold to $\tau = 0.85$ to accommodate environmental noise. Training on the real robot used LoRA-based fine-tuning for 50,000 steps, and all evaluations were performed on an NVIDIA RTX 4090 GPU.

# E   More Experiment Results

## E.1   Simulation Experiment

**LIBERO Task Definitions.**   Similarly, we also utilize all task suites provided in LIBERO for our evaluations. The Robosuite-based robot setup includes the following tasks: 1) "place bowl on plate with spatial variation" (e.g., drawer positions), 2) "pick object" (e.g., ketchup, bowl, apple), 3) "(open / close) target drawer; action object" (e.g., "open top drawer; place apple into drawer"), and 4) "achieve goal using shared objects" (e.g., rearranging spatial relationships or altering object states).

**SIMPLER Task Definitions.**   We utilize all task variants provided in SIMPLER for our evaluations, which include the Google robot setup with the following tasks: 1) "pick Coke can", 2) "move obj1 near obj2", 3) "(open / close) (top / middle / bottom) drawer", and 4) "open top drawer; place apple into top drawer". Evaluations for the Google robot setup are provided for both Visual Matching (VM) and Variant Aggregations (VA).

**Implementation Details.**   Simulated evaluations for CogACT and SIMPLER are conducted on a single NVIDIA RTX 4090 GPU in BF16 precision. During inference, we use DDIM sampling with 10 steps and a classifier-free guidance (CFG) coefficient of 1.5. Similarly, for OpenVLA and LIBERO, inference is performed on a single NVIDIA RTX 4090 GPU in BF16 precision.

## E.2   Additional Simulation Results

**Result of Subtask on LIBERO Spatial Task Suit.**   Table 6 presents detailed results on each subtask in the LIBERO-Spatial suite. We observe that VLA-Cache, along with methods like SparseVLM and FastV, occasionally surpasses the baseline's success rate on individual subtasks. This suggests that certain redundant tokens may distract the baseline model, and pruning or reusing tokens can in fact enhance its robustness.

**Visualization Results.**   Figure 5 present evaluation examples of each tasks executed by OpenVLA with VLA-Cache in LIBERO. More visualized results are available in the supplementary materials.

Table 6: Comparison of success rates across different tasks in the LIBERO-Spatial benchmark.

| Method | Success Rate % ↑ | | | | | | | | | | | FLOPs ↓ | CUDA Time (ms) ↓ |
|---|---|---|---|---|---|---|---|---|---|---|---|---|---|
| | 1 | 2 | 3 | 4 | 5 | 6 | 7 | 8 | 9 | 10 | Avg | | |
| Baseline (OpenVLA) | 90 | 90 | 84 | 96 | 70 | 90 | 96 | 76 | 82 | 70 | 84.4 | 1.888 | 52.37 |
| SparseVLM | 88 | 60 | 90 | 90 | 60 | 82 | 90 | **92** | 72 | **74** | 79.8 | 1.367 | 88.08 |
| FastV | **92** | 90 | **90** | 94 | 58 | **92** | 90 | 80 | **78** | 70 | 83.4 | 1.888 | 54.00 |
| **VLA-Cache** | 90 | **90** | 88 | **94** | **66** | 84 | **94** | 84 | 76 | 72 | **83.8** | **1.382** | **32.22** |

Table 7: Real-world *PickPot* task under dynamic background (human/object motion).

| Method | Success (%) ↑ | FLOPs ↓ | Latency (ms) ↓ |
|---|---|---|---|
| Baseline (OpenVLA) | 95 | 1.800 | 68.03 |
| + Noise | 80 | 1.807 | 68.22 |
| + Noise, VLA-Cache | 80 | **1.275** | **50.59** |

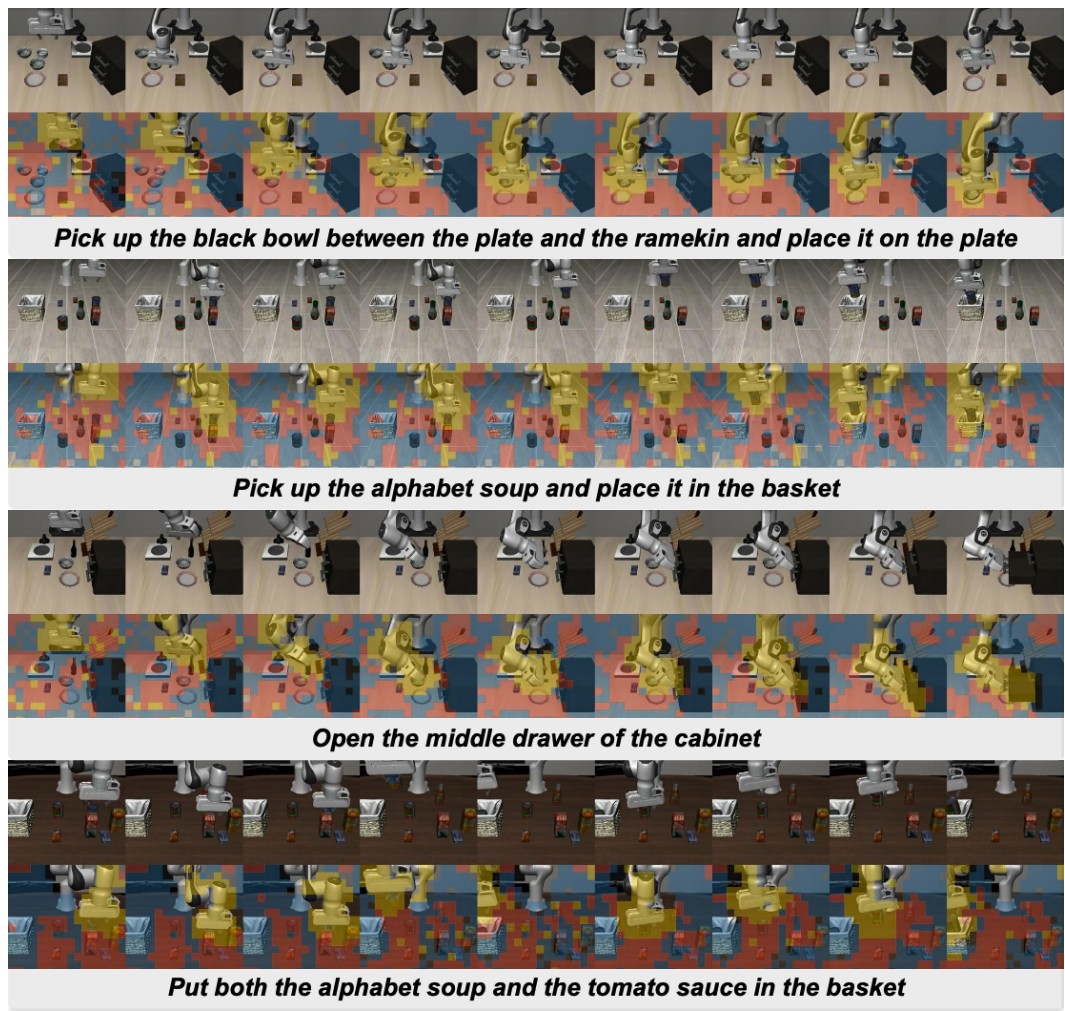

*Pick up the black bowl between the plate and the ramekin and place it on the plate*

*Pick up the alphabet soup and place it in the basket*

*Open the middle drawer of the cabinet*

*Put both the alphabet soup and the tomato sauce in the basket*

Figure 5: VLA-Cache test results and attention heat map in a simulated environment

## E.3 Additional Ablations and Comparisons

**Attention vs. object–mask proxies for task relevance.** To validate whether attention score is a reliable proxy for task relevance, we compared our default attention–based filtering with an object-

mask alternative using *Efficient Track Anything*[39]. On LIBERO-Spatial with OpenVLA-OFT, attention-based proxy yields both higher success rate and lower latency than the mask variant:

| Method | SR (%) ↑ | FLOPs(T) ↓ | Latency (ms) ↓ | Freq. (Hz) ↑ |
|---|---|---|---|---|
| OPENVLA-OFT | 97.8 | 3.99 | 78.35 | 65.44 |
| + VLA-Cache (Attention) | **98.3** | **3.04** | **61.12** | **81.67** |
| + VLA-Cache (Object Mask) | 87.4 | 3.15 | 87.49 | 64.78 |

Table 8: Attention vs. object-mask proxies for task relevance on LIBERO-SPATIAL.

While object masks provide spatial localization, they can miss fine-grained or contextual signals essential for manipulation, especially with background clutter or small, relevant parts. In contrast, attention scores are dynamically produced and tightly coupled with the model's internal reasoning, providing a lightweight, task-adaptive proxy for relevance.

**Sensitivity to static-token budget $k$ and relevance threshold $\tau$.** We further analyze the trade-off between accuracy and efficiency by varying the number of static tokens $k$ (with $\tau=0.5$ fixed) and the task-relevance threshold $\tau$ (with $k=100$ fixed) on LIBERO-Spatial using OpenVLA-OFT. Results indicate robustness across a wide range; our default ($k=100$, $\tau=0.5$) offers a strong balance.

| Method | SR (%) ↑ | FLOPs (T) ↓ | Latency (ms) ↓ | Freq. (Hz) ↑ |
|---|---|---|---|---|
| OPENVLA-OFT | 97.8 | 3.995 | 78.35 | 65.44 |
| + VLA-Cache ($k=50$) | 97.6 | 3.332 | 66.82 | 77.23 |
| + VLA-Cache ($k=80$) | 97.8 | 3.226 | 66.55 | 78.66 |
| + VLA-Cache ($k=100$) | **98.2** | 3.156 | 64.88 | 79.88 |
| + VLA-Cache ($k=120$) | 98.0 | 3.109 | 62.99 | 80.56 |
| + VLA-Cache ($k=150$) | 97.4 | **3.043** | **61.12** | **81.67** |
| + VLA-Cache ($k=180$) | 96.6 | 2.936 | 60.46 | 82.51 |

Table 9: Varying the static-token budget $k$ (with $\tau=0.5$).

| Method | SR (%) ↑ | FLOPs (T) ↓ | Latency (ms) ↓ | Freq. (Hz) ↑ |
|---|---|---|---|---|
| OPENVLA-OFT | 97.8 | 3.995 | 78.35 | 65.44 |
| + VLA-Cache ($\tau=0.2$) | 95.6 | 3.384 | 66.93 | 76.74 |
| + VLA-Cache ($\tau=0.3$) | 96.2 | 3.283 | 67.03 | 79.31 |
| + VLA-Cache ($\tau=0.4$) | 98.0 | 3.204 | 66.38 | 79.79 |
| + VLA-Cache ($\tau=0.5$) | **98.2** | 3.156 | 64.88 | 79.88 |
| + VLA-Cache ($\tau=0.6$) | 98.6 | 3.131 | 64.27 | 81.98 |
| + VLA-Cache ($\tau=0.7$) | 98.4 | **3.068** | **63.12** | **82.96** |

Table 10: Varying the relevance threshold $\tau$ (with $k=100$).

Overall, efficiency (FLOPs and latency) improves monotonically with larger $k$ and $\tau$, while success rate remains consistently high, corroborating the stability of *VLA-Cache* across sensitivity settings. Our default ($k=100$, $\tau=0.5$) is used for all main results unless stated otherwise.

**Applicability to diffusion-based and alternative VLA architectures.** *VLA-Cache* is designed to accelerate the language-decoder stage of VLAs and is therefore not directly applicable to models without a vision–language backbone, such as standalone diffusion policies. However, for hybrid architectures that combine a VLM with a diffusion-based policy head (e.g., CogACT [19]), our method remains fully compatible and has shown consistent efficiency gains and stable task performance in the SIMPLER environment, as shown in Table 3. Since *VLA-Cache* operates by reusing temporally static visual tokens before action generation, it can be integrated with diffusion-based or transformer-style models (e.g., RDT, DiT) that exhibit inter-frame redundancy, offering a promising direction for future generalization.

### E.4 Real Robot Experiment

**Robot Setup.** The setup of the Franka Robot is shown in Figure 6. In this example, a Kinova Jaco robot arm with 6 degrees of freedom is rigidly fixed to the frame. We use a Sony AX53 camera, which is placed opposite the robot arm. The camera is facing the operating table and transmits the video in real time.

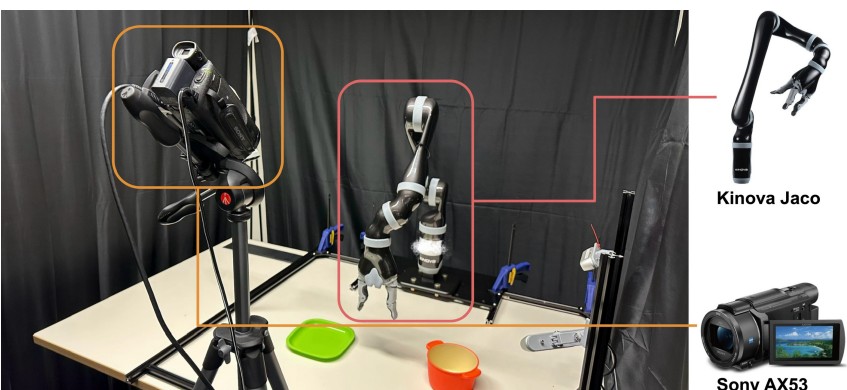

Figure 6: Kinova Jaco Robot Setup

**Data Collection.** Our data collection work is based on the *CLVR_Jaco_Play* dataset. We employed PyBullet as an inverse kinematics (IK) controller. The system receives incremental Cartesian displacement inputs $(\Delta x, \Delta y, \Delta z)$ from an Xbox controller, which are processed to generate joint velocity commands. These commands are transmitted to the robotic arm's velocity controller at a 10Hz control frequency for real-time execution. We record four types of observations: Third-person camera observations (**front_cam_ob**), End-effector Cartesian pose (**ee_cartesian_pos_ob**), End-effector Cartesian velocity (**ee_cartesian_vel_ob**), Jaco arm joint positions (**joint_pos_ob**).

**Data Preprocessing.** The preprocessing procedure was conducted as follows. Initially, the recorded frames underwent center cropping, reducing the resolution from 1280×720 to 912×720, followed by resizing to 224×224. Subsequently, episodes were manually selected based on a visual inspection of the video sequences generated from the recorded frames. To minimize potential biases associated with excessively long episodes, those exceeding 250 steps were excluded from the dataset. Furthermore, steps in which all recorded action values were zero were removed to ensure data relevance and integrity.

**Task Definitions and Success Criteria.** We design four single-instruction tasks for the real-world robot setting to evaluate manipulation performance across diverse object types and interaction modes. The task definitions are as follows:

1. **Pick Up Orange Pot** (PickPot): The robot's goal is to grasp the orange pot and lift it completely off the table. A trial is successful if the pot is grasped and raised with visible clearance from the surface. We collected 218 valid demonstrations for the training dataset, with slight random adjustments to the initial positions of the pot and the robotic arm in each episode. During evaluation, each trial is recorded as a success (1) or failure (0); there is no partial credit.

2. **Place Blue Cube in Box** (PlaceCube): The robot's goal is to place the held blue cube into the target container. A trial is successful if the cube is fully inside the container at the terminal step without contact-induced ejection. We collected 212 valid demonstrations for the training dataset, with randomized container positions and robotic arm initial configurations in each episode. During evaluation, each trial is recorded as a success (1) or failure (0); there is no partial credit.

3. **Put Sausage in Blue Pan** (PutSausage): The robot's goal is to stably place the held sausage into the blue pan. Success requires the sausage to rest inside the pan boundary at the terminal step. We collected 219 valid demonstrations for the training dataset, with randomized pan

coordinates and robotic arm joint angles across trials. During evaluation, each trial is recorded as a success (1) or failure (0); there is no partial credit.

4. **Wipe Table** (WipeTable): The robotic arm's goal is to sweep scattered items into a fixed-position dustpan using a broom. Success requires all designated items to be inside the dustpan area at termination. For the training dataset, we collected 187 valid demonstrations featuring randomized placements of simulated items (e.g., fries/cheese) on the table and varied initial joint configurations of the robotic arm, while the dustpan location remained fixed. This task explicitly incorporates dynamic environmental variations to rigorously test generalization capabilities. During evaluation, each trial is recorded as a success (1) or failure (0); there is no partial credit.

**Trial Protocol and Randomization.** We run **20** trials for *PickPot* and *PutSausage*, and **30** trials for *PlaceCube* and *WipeTable*, for a total of **100** trials per method. For each trial we randomize the initial robot configuration and the object placement within a bounded region of the workspace. Trials are single-attempt with a fixed horizon; there is no human intervention or reset within a trial. Outcomes are recorded as binary success or failure per the criteria above.

| Task | OPENVLA | | | OPENVLA + VLA-CACHE | | |
| --- | --- | --- | --- | --- | --- | --- |
| | Success | Failure | SR (%) | Success | Failure | SR (%) |
| PickPot (20) | 19 | 1 | 95.0 | 18 | 2 | 90.0 |
| PlaceCube (30) | 25 | 5 | 83.3 | 27 | 3 | 90.0 |
| PutSausage (20) | 16 | 4 | 80.0 | 17 | 3 | 85.0 |
| WipeTable (30) | 21 | 9 | 70.0 | 22 | 8 | 73.3 |
| **Total (100)** | 81 | 19 | 82.1 | 84 | 16 | 84.6 |

Table 11: Real-world results with trial counts and success rates.

**Results with Counts and Rates.** Table 11 reports per-task successes and failures, along with the corresponding success rate. Average success is computed across all 100 trials.

**Visualization Results.** Figure 7 present evaluation examples of each tasks executed by OpenVLA with VLA-Cache on the Kinova Jaco Robot Arm. More visualized results are available in the supplementary materials.

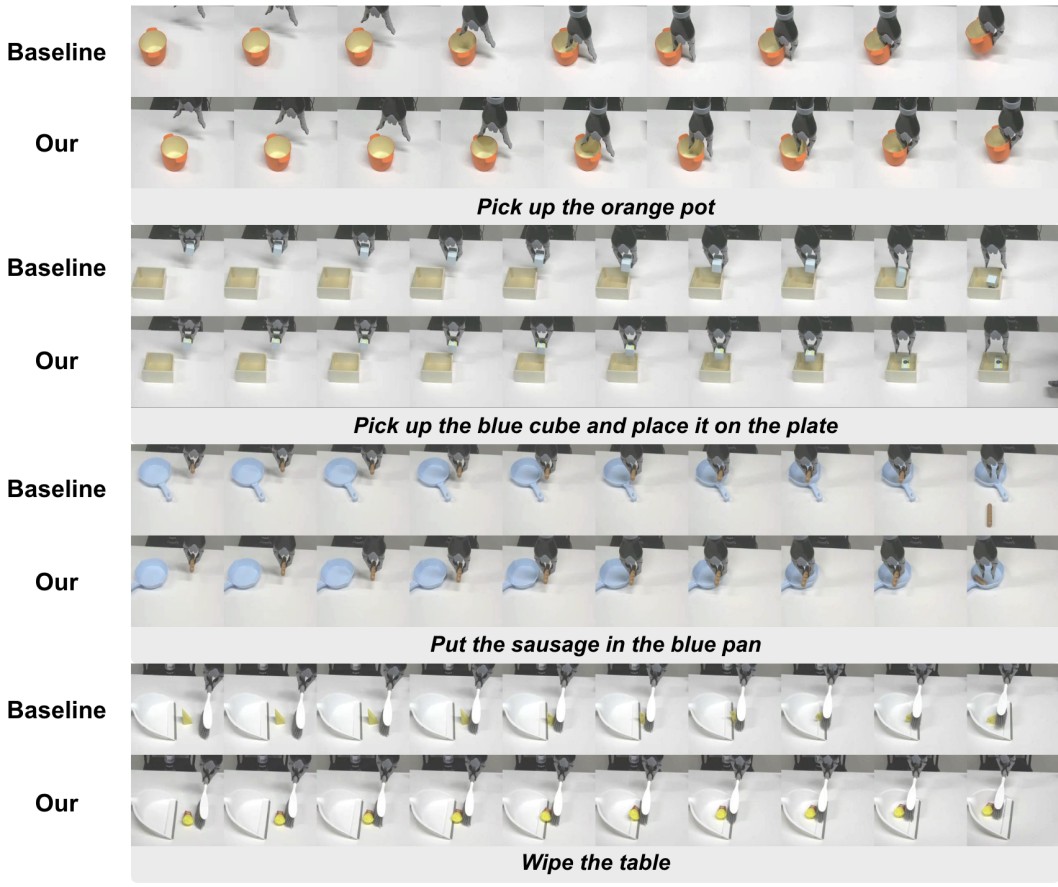

Figure 7: Comparison between baseline (OpenVLA) and VLA-Cache in real environment.

