# OpenReview forum: "VLA-Cache: Efficient Vision-Language-Action Manipulation via Adaptive Token Caching"
_NeurIPS.cc/2025/Conference — NeurIPS 2025 poster_

### Official Review · Reviewer_zsdX · 2025-07-02

**Clarity:** 2
**Significance:** 2
**Originality:** 2
**Rating:** 4
**Confidence:** 3

**Summary:**

The paper proposes VLA-Cache, which is a training-free inference acceleration method that can help speedup for robotic control. VLA-Cache selects visually static tokens and task-relevant tokens to construct reusable tokens, and reuse KV-cache for better computational efficiency.  Additional techniques like layer adaptive reusing is also added to improve the performance. The authors evaluate the method on several benchmarks to demonstrate the improvement on the inference speed.

**Questions:**

1.	For calculating the task-relevant tokens, what visual tokens are you used to calculate the attention matrix? It seems that we need visual tokens at different layers to do this, which is the target for reduction. Do you use the raw image tokens just like OpenVLA or do you use the attention score from the last step?

2.	I think the main idea is try to reuse the tokens that won’t cast too much influence on the final tokens. It would be interesting to see the performance of a variant which doesn’t calculate visually static and re-use all tokens that belong to Full set – task relevant tokens.

3.	For wrist camera used in OpenVLA-OFT (L284), can you explain how to handle it? How many tokens can be reused? It seems like most tokens will not be visually static anymore.

4.	For control frequency result in Table 2/3/5, most of the improvements seem to be not prominent (+0.36Hz / 0.2Hz for OpenVLA, +2.2Hz for CogACT). How many tokens are reused? Why is the case?

**Ethical Concerns:**

["NO or VERY MINOR ethics concerns only"]

**Final Justification:**

The method tries to accelerate the inference and the method seems to be novel. Though the performance gain seems limited to certain scenarios, I think the method itself seems to be novel.

**Quality:**

2

**Strengths And Weaknesses:**

Strengths:

-	In general, the paper is clear and easy to follow

-	The idea of adaptively reusing the KV cache of certain regions is interesting and seems to be novel.

Weaknesses:

-	Although the method can improve theoretical FLOPs and CUDA latency, the improvement on actual control frequency seems to be limited.

-	See questions below.

---

> ### Author Rebuttal · Authors · 2025-07-31
>
> We sincerely thank you for your insightful comments and constructive feedback. Below, we address your comments and questions in detail.
>
> ---
>
> **Q1. Which visual tokens are used to compute attention?**
>
> **R1:** Thanks for the question. We compute task relevance scores using the decoder self-attention weights from the previous timestep (t−1). We do not access or recompute visual tokens from multiple encoder layers, nor do we require raw image tokens as in OpenVLA. Instead, we **reuse existing attention weights that were already computed during action prediction at the previous timestep**.
>
> ---
>
> **Q2. Experiment: Reusing only task-relevant tokens (no static filtering)**
>
> **R2:** Our method is designed to reuse static tokens, those unlikely to influence final predictions, to reduce redundant computation without harming performance. Following your suggestion, we conducted an experiment on the LIBERO Spatial benchmark where only task-relevant tokens were reused, and all static tokens were recomputed. The results are shown below:
>
> | Method                     | SR (%) | FLOPs ↓ | Latency (ms) ↓ | Freq. (Hz) ↑ | Ratio (Primary) | Ratio (Wrist) |
> |-----|-------|--------|-------|-------|--------|---|
> | OpenVLA-OFT                | 97.8   | 3.995    | 78.35          | 65.44         | -                | -               |
> | + VLA-Cache                | 98.4   | 3.043    | 61.12          | 81.67         | 31.62            | 35.47           |
> | + VLA-Cache (task-only)    | 0.0    | 2.912    | 59.49          | 84.78         | 37.67            | 36.58           |
>
> As expected, reusing all task-relevant tokens leads to complete performance failure (SR = 0%), confirming that these tokens are essential for accurate action prediction and should not be reused. In contrast, VLA-Cache reuses only static tokens and preserves task-relevant computations, thus maintaining high success rates while improving efficiency.
>
> ---
>
> **Q3. Token reuse for wrist camera input**
>
> **R3:**
> Thank you for the thoughtful question. The wrist camera input is processed identically to the third-person camera in our system. While it is true that wrist-view observations contain more motion (due to hand-object interaction and egocentric perspective shifts), not all tokens are equally dynamic. Our top-$k$ similarity-based selection captures local temporal consistency. For example, background areas, parts of the robot arm, or regions with limited motion—still exhibit high similarity across frames. These static regions enable effective token reuse.
>
> As shown in Q2-R2, the token reuse ratios are surprisingly similar between third-person view (31.62%) and wrist view (35.47%). This indicates that, despite the higher motion, the wrist camera input still contains substantial redundancy that VLA-Cache can exploit for acceleration without hurting performance.
>
> ---
>
> **Q4. Control Frequency Improvement**
>
> **R4:**
> Thanks for the nice question. Across OpenVLA, OpenVLA-OFT, and CogACT, we apply the same configuration. Token reuse ratios are consistently around 35% as reported above (Q2-R2).
>
> VLA-Cache primarily accelerates the language decoder, which accounts for over 90% of total compute in VLA models (based on FLOPs profiling). As such, control frequency improvements scale with how often the decoder is called. In low-frequency models like OpenVLA (≈4Hz), which predict one action per timestep, the performance gain is naturally limited, even a large reduction in decoder latency translates to only ~0.2–0.4Hz gain, due to other system bottlenecks like policy update frequency or observation preprocessing. In contrast, high-frequency models like OpenVLA-OFT (≈65Hz), which use action chunking to predict multiple actions per step, benefit much more. For example, we observe a +13.88Hz gain in Table 2.
>
> ---
>
> **We sincerely appreciate your valuable suggestions and hope our rebuttal can address your concerns. We are happy to answer any remaining questions during the discussion phase.**

---

> > ### Comment · Reviewer_zsdX · 2025-08-04
> >
> > I thank the authors for the response. Most of my previous questions are clarified. I would like to maintain the positive score.

---

> > > ### Author Response · Authors · 2025-08-04
> > >
> > > Thank you for your kind follow-up and for acknowledging our rebuttal. We truly appreciate your valuable feedback throughout the review process.
> > >
> > > If there are any remaining concerns or questions you would like us to elaborate on, we would be more than happy to provide further clarification during the discussion phase.
> > >
> > > Thank you again for your constructive feedback.

---

### Official Review · Reviewer_B8wU · 2025-07-03

**Clarity:** 2
**Significance:** 2
**Originality:** 2
**Rating:** 4
**Confidence:** 4

**Summary:**

This paper introduces VLA-Cache, a training-free inference acceleration method for Vision-Language-Action (VLA) models.
The key contributions are: 1) Static Token Selection. 2) Evicting Task-Relevant Tokens. 3) Layer Adaptive Token Reusing.
The method is evaluated on two simulation platforms (LIBERO and SIMPLER) and a real-world robotic setting.
Results demonstrate that VLA-Cache achieves a speedup of up to 1.7× in CUDA latency and a 15% increase in control frequency, with negligible loss in task success rate. The paper also shows the real-world applicability of VLA-Cache by deploying it on a Kinova Jaco2 robot arm.

**Questions:**

See above.

**Ethical Concerns:**

["NO or VERY MINOR ethics concerns only"]

**Final Justification:**

Thank you to the authors for their detailed rebuttal. Your responses have satisfactorily addressed most of my concerns, and as a result, I have raised my score to a 4.

**Limitations:**

YES

**Quality:**

3

**Strengths And Weaknesses:**

Strengths:

1. The paper is well-organized and features a clear and comprehensible writing style. The author presents the motivation for the proposed method clearly and understandably. Subsequently, each component of the method, together with the experimental settings and results, is elaborated in detail.

2. This paper introduces a novel training-free inference acceleration method, comprising three key components: Static Token Selection, Evicting Task-Relevant Tokens, and Layer-Adaptive Token Reusing.

3. The paper establishes and conducts extensive experiments across three benchmarks: LIBERO, SIMPLER, and real-robot. These experiments serve to demonstrate the effectiveness of the proposed method.


Weaknesses:

1. The thresholds and k in Equations (5),(7),(9) need to be set carefully to better retain stable tokens. Otherwise, important visual areas may be misclassified as static regions, or static regions may be wrongly regarded as task-relevant ones for action prediction.

2. Calculating the attention matrix for Evicting Task-Relevant Tokens involves computing K, Q, V. It is unclear whether KV-cache is used here. If it is used, this might lead to inaccuracies in calculating the attention matrix. If not, this would increase the computational load for KV.

3. It is not clear whether this method is applicable to diffusion-based VLA models.

4. While the method reduces FLOPs, Latency, and improves Control Frequency while maintaining task success rates across LIBERO, SIMPLER, and real-robot experiments, the overall performance improvement appears to be limited. For instance, in the real-robot experiment, the success rate for the PickPot task decreased from 95% to 90%, while the Control Frequency increased only slightly from 4.02 to 4.21.

---

> ### Author Rebuttal · Authors · 2025-07-31
>
> The authors sincerely than you for your insightful comments and constructive feedback. Below, we address your comments and questions in detail.
>
> ---
>
> **Q1. Sensitivity of Thresholds (k) and ($\tau$)**
>
> **R1:** The parameters \(k\) (for static token selection) and \($\tau$\) (for task relevance thresholding) can influence the trade-off between accuracy and efficiency. To further examine the sensitivity of these parameters, we conducted additional experiments on the LIBERO Spatial benchmark using the OpenVLA-OFT model. The results are summarized below.
>
> *Table 1. Varying \(k\) values (with $\tau$ = 0.5):*
>
> | Method                     | SR (%) | FLOPs ↓ | Latency (ms) ↓ | Freq. (Hz) ↑ |
> |----------------------------|--------|----------|----------------|---------------|
> | OpenVLA-OFT                | 97.8   | 3.995    | 78.35          | 65.44         |
> | + VLA-Cache (k=50)         | 97.6   | 3.332    | 66.82          | 77.23         |
> | + VLA-Cache (k=80)         | 97.8   | 3.226    | 66.55          | 78.66         |
> | + VLA-Cache **(k=100)**        | 98.2   | 3.156    | 64.88          | 79.88         |
> | + VLA-Cache (k=120)        | 98.0   | 3.109    | 62.99          | 80.56         |
> | + VLA-Cache (k=150)        | 97.4   | 3.043    | 61.12          | 81.67         |
> | + VLA-Cache (k=180)        | 96.6   | 2.936    | 60.46          | 82.51         |
>
>
> *Table 2. Varying $\tau$ values (with k = 100):*
>
> | Method                     | SR (%) | FLOPs ↓ | Latency (ms) ↓ | Freq. (Hz) ↑ |
> |----------------------------|--------|----------|----------------|---------------|
> | OpenVLA-OFT                | 97.8   | 3.995    | 78.35          | 65.44         |
> | + VLA-Cache (τ=0.2)        | 95.6   | 3.384    | 66.93          | 76.74         |
> | + VLA-Cache (τ=0.3)        | 96.2   | 3.283    | 67.03          | 79.31         |
> | + VLA-Cache (τ=0.4)        | 98.0   | 3.204    | 66.38          | 79.79         |
> | + VLA-Cache **(τ=0.5)**        | 98.2   | 3.156    | 64.88          | 79.88         |
> | + VLA-Cache (τ=0.6)        | 98.6   | 3.131    | 64.27          | 81.98         |
> | + VLA-Cache (τ=0.7)        | 98.4   | 3.068    | 63.12          | 82.96         |
>
> These results show that VLA-Cache is generally robust to changes in \(k\) and \($\tau$\). While efficiency (FLOPs and latency) improves with higher values, task success rate remains consistently high across settings. Our default configuration (k = 100, τ = 0.5) offers a strong trade-off between performance and efficiency.
>
> ---
>
> **Q2. Whether Task Relevance Computation Uses KV-Cache**
>
> **R2:**
> Thanks for the question. We do not recompute $Q$, $K$ and $V$ values during inference for task-relevant token selection. Instead, we reuse the attention scores from the decoder at the previous timestep (t−1). These attention scores were already computed for action prediction and reflect semantically meaningful task-driven importance.
>
> To validate this choice, we conducted an ablation comparing:
>
> * _Current Implementation_: Using full decoder attention from the previous timestep $t−1$;
>
> * _A baseline variant_ that recomputes $Q$, $K$ and $V$ for the first two decoder layers at timestep $t$, and uses their attention scores.
>
> | Method  | SR (%) | FLOPs ↓ | Latency (ms) ↓ | Freq. (Hz) ↑ |
> |-----|------|----|------|----|
> | OpenVLA-OFT  | 97.8 | 3.995 | 78.35 | 65.44 |
> | + VLA-Cache (Current implementation)| **98.4** | **3.043** | **61.12** | **81.67** |
> | + VLA-Cache (Variant)  | 96.6 | 3.371 | 68.39 | 76.64 |
>
> The results show that our method achieves both better efficiency and higher task success. Recomputing shallow attention at the current timestep leads to worse SR and latency, likely due to insufficient task-level semantics in early decoder layers. This supports our design choice of reusing full attention from the prior timestep.
>
> ---
>
> **Q3. Applicability to Diffusion-Based VLA Models**
>
> **R3:**
> We would like to clarify that **our paper already includes experiments on CogACT [1], a representative diffusion-based VLA model**. Specifically, Table 3 in our main paper reports the performance of VLA-Cache applied to CogACT in the SIMPLER environment. These results demonstrate that our method is compatible with diffusion-based VLA models and remains effective in this setting.
>
> VLA-Cache operates at the VLM decoder level by identifying and reusing static visual tokens across time. This design is agnostic to the decoder architecture and naturally integrates with diffusion-based policies like CogACT. Since token caching occurs before action generation, it does not interfere with the diffusion process.
>
> As shown in our results, applying VLA-Cache to CogACT improves inference efficiency (e.g., lower FLOPs and latency) without compromising task success, confirming both compatibility and benefit.
>
> *Table3: Comparison of VLA-Cache with CogACT in SIMPLER Environment*
>
> | SIMPLER      | Method              | PickCan | MoveNear | Drawer | DrawerApple | Avg.  | FLOPs (T) ↓ | Latency (ms) ↓ | Freq. (Hz) ↑ |
> |--------------|---------------------|---------|----------|--------|--------------|-------|--------------|----------------|----------------|
> | Matching     | CogACT              | 91.3%   | **85.0%**| **71.8%** | 50.9%        | **74.8%** | 1.847        | 54.29          | 12.42          |
> |              | + **VLA-Cache**     | **92.0%** | 83.3%    | 70.5%  | **51.6%**     | 74.4%  | **1.496**    | **39.63**      | **14.66**      |
> | Aggregation  | CogACT              | 89.6%   | 80.8%    | 28.3%  | **46.6%**     | 61.3%  | 1.807        | 53.54          | 12.36          |
> |              | + **VLA-Cache**     | **91.7%** | **79.3%**| **32.5%** | 45.8%        | **62.3%** | **1.493**    | **39.11**      | **14.48**      |
>
>
> **Reference:**
>
> [1] Li, Q., et al. (2024). *Cogact: A foundational vision-language-action model for synergizing cognition and action in robotic manipulation*. arXiv:2411.19650
>
> ---
>
> **Q4. Performance Drop in PickPot and Limited Control Frequency Improvement**
>
> **R4:**
> Thanks for the nice concern. We acknowledge that the PickPot task shows a 5% drop in real-robot success rate. However, this change is within the expected statistical variance, given only 20 trials per task. Other real-world tasks such as PlaceCube show significant improvements (e.g., +7.3%), indicating no systematic degradation from applying VLA-Cache. Importantly, VLA-Cache is designed to improve computational efficiency while maintaining performance. Across all evaluated domains (LIBERO, SIMPLER, real-robot), task success rates remain stable or improve, confirming that caching does not harm control quality.
>
> Regarding the improvement in control frequency, we note that our method accelerates the language decoder, which accounts for most of the VLA model’s computation. The end-to-end gain scales with decoder usage: high-frequency models like OpenVLA-OFT see substantial improvements (+13.88 Hz), while low-frequency pipelines (e.g., OpenVLA real-robot) are also constrained by external system bottlenecks such as sensor and actuator delays. Thus, the modest frequency gain in PickPot reflects platform-level limits rather than algorithmic shortcomings.
>
> ---
>
> **We sincerely appreciate your valuable suggestions and hope our rebuttal can address your concerns. We are happy to answer any remaining questions during the discussion phase.**

---

> > ### Comment · Reviewer_B8wU · 2025-08-07
> >
> > Thank you to the authors for their detailed rebuttal. Your responses have satisfactorily addressed most of my concerns, and as a result, I have raised my score to a 4.

---

> ### Author Response · Authors · 2025-08-06
>
> We sincerely thank you for your efforts in reviewing our paper. We have provided corresponding responses and results, which we believe have covered your concerns. We hope to further discuss with you whether your concerns have been addressed or not. Please let us know if you still have any unclear parts of our work.
>
> Best Regards,
>
> Authors of Submission 5408

---

### Official Review · Reviewer_bXk4 · 2025-07-03

**Clarity:** 3
**Significance:** 2
**Originality:** 2
**Rating:** 4
**Confidence:** 5

**Summary:**

This paper proposes VLA-Cache, an inference-time acceleration method designed to improve the efficiency of Vision-Language-Action (VLA) models. The core idea leverages the spatial redundancy of static regions across visual frames, reusing representations of static tokens via a key-value (KV) caching mechanism to reduce redundant computation. The authors also introduce an attention-based filtering strategy and a layer-adaptive reuse policy to maintain semantic accuracy of key tokens.

**Questions:**

Add at least one real-world robotic experiment, even on a low-cost platform, to demonstrate practical applicability and robustness.

Extend validation to a wider range of VLA backbones, such as Gemma, Qwen-VL, or CogAct, to assess generality and compatibility.

Analyze the limitations of using attention scores as a proxy for semantic relevance. Consider introducing alternative signals such as object masks or affordance maps.

Explore combinations with other token optimization strategies, such as token merging or patch dropout, to enhance performance further.

**Ethical Concerns:**

["NO or VERY MINOR ethics concerns only"]

**Final Justification:**

Sorry for the late response to the final justification, as had a serious cold in recent days. As I claimed in the review and the rebuttal, I had raised my final rating to 4, as the author solved most of my concerns.

**Limitations:**

yes

**Paper Formatting Concerns:**

good

**Quality:**

2

**Strengths And Weaknesses:**

# Strengths
Clear innovation: Proposes a plug-and-play token reuse strategy that requires no retraining, making it lightweight and practical.
Well-designed method: Clever integration of visual staticness and semantic importance avoids performance degradation due to over-caching.

Comprehensive evaluation metrics: Includes analysis on FLOPs, CUDA latency, control frequency, and task success rate.

Strong baselines and comparisons: Systematic comparison with multiple token-level acceleration baselines such as FastV and SparseVLM.
# Major Weaknesses
- No real-world robotic experiments: The method is only validated in simulation environments, with no deployment or testing on real robots, raising concerns about its real-world feasibility and robustness.
- Limited validation scope: The method is only tested on a single visual backbone (e.g., LLaMA2), lacking generalization experiments on other architectures (e.g., Gemma, Qwen-VL, CogVideo).
- Reliance on attention as a proxy for task relevance is heuristic in nature, and no alternatives or robustness analyses are provided.
- Contribution is local rather than fundamental: While practical, the method is more of an engineering optimization and offers limited insight into the learning mechanisms of VLA models.

---

> ### Author Rebuttal · Authors · 2025-07-31
>
> We sincerely appreciate your constructive and detailed feedback. Below, we address each concern point-by-point.
>
> ---
>
> **Q1&W1. Lack of Real-World Robot Experiments**
>
> **R1:**
> Thank you for the suggestion. We would like to clarify that **VLA-Cache has already been deployed and evaluated on a real robotic platform**, the **Kinova Jaco2** arm. The real-world results are thoroughly reported in Section 5.2, Section 5.4, and Appendix E.3, with quantitative performance summarized in Table 5, and qualitative outcomes illustrated in Figure 3 (right) and Figure 4b. Across four real-world manipulation tasks, VLA-Cache improved **success rate** by **2.4%**  on average, reduced **FLOPs** by **28.2%**, and lowered **latency** by **19.2%** compared to the OpenVLA baseline.
>
> In addition, we conducted experiments under **dynamic real-world disturbances** such as human motion and background clutter to further assess robustness, as detailed in **Section 5.4** and **Table 7**. The method maintained a high success rate while significantly reducing FLOPs and latency.
>
> Furthermore, demonstration videos of the real-robot execution has been provided via **anonymous link in the abstract**, showcasing the method’s practical applicability in real-world settings.
>
>
> ---
>
> **Q2&W2. Limited Backbone Generalization**
>
> **R2:**
> Thanks for the nice question. We would like to clarify that besides OpenVLA, we have already evaluated VLA-Cache across more diverse and representative VLA architectures (i.e., **CogACT** and OpenVLA-OFT). Notably, CogACT is a diffusion-based VLA model, which differs structurally from standard autoregressive LLM decoders, and our results on it are reported in **Table 3**. On the Visual Aggregation task, VLA-Cache improved success rate by **+1.0%**, reduced FLOPs by **17.4%**, latency by **27.0%**, and increased control frequency by **+2.12 Hz**.
>
> Moreover, most recent state-of-the-art VLA systems (e.g., OpenVLA, CogACT, OpenVLA-OFT, UniVLA) are all built upon **LLaMA2** as the language decoder. Since our current evaluation already covers this dominant backbone family across multiple variants (autoregressive, diffusion-based, and OFT-based), we believe the generalization ability of VLA-Cache is well demonstrated.
>
>
> ---
>
> **Q3&W3. Limitations of Using Attention scores as Task-Relevance Proxies**
>
> **R3:**
> Thank you for the constructive suggestion. We acknowledge that attention scores may not perfectly align with human-understood semantics.
> To assess the reliability of attention scores as a proxy for semantic relevance, we conducted a direct comparison between our **attention-based** token filtering and an **object-mask-based** approach using **Efficient Track Anything** [1,2] on the LIBERO Spatial benchmark. The results are shown in the following table.
>
> | Method                              | SR (%) | FLOPs ↓ | Latency (ms) ↓ | Freq. (Hz) ↑ |
> |-------------------------------------|--------|----------|----------------|---------------|
> | OpenVLA-OFT                         | 97.8   | 3.995    | 78.35          | 65.44         |
> | + VLA-Cache (attention score)       | **98.3**   | **3.043**    | **61.12**          | **81.67**         |
> | + VLA-Cache (object mask)     | 87.4   | 3.157    | 87.49          | 64.78         |
>
> We observe that attention-based filtering not only improves accuracy (more than 10%) but also reduces latency and computational cost, compared to the object-mask variant. While object masks provide spatial localization, they often miss fine-grained or contextual signals essential for manipulation tasks, especially when background clutter or small, relevant parts might be involved. In contrast, attention scores are dynamically generated and tightly coupled with the model’s internal reasoning, providing a lightweight, task-adaptive proxy for relevance.
>
> Due to this year’s rebuttal policy, we cannot include visualizations, but we will incorporate illustrative examples in the final version to better explain this behavior.
>
> **Reference:**
>
> [1] Xiong, Y., et al. (2024). *Efficient Track Anything*. arXiv:2411.18933
> [2] Xiong, Y., et al. (2024). *EfficientSAM: Leveraged Masked Image Pretraining for Efficient Segment Anything*. CVPR 2024
>
> ---
>
> **Q4. Compatibility with Token Merging and Pruning Strategies**
>
> **R4:**
> Thanks for the insightful comment. To validate this compatibility, we implemented and evaluated VLA-Cache in combination with token merging (SparseVLM [3]) and token pruning (FastV [4]) methods. The results are summarized below:
>
> | Method                              | SR (%) | FLOPs ↓ | Latency (ms) ↓ | Freq. (Hz) ↑ |
> |-------------------------------------|--------|----------|----------------|---------------|
> | OpenVLA-OFT                         | 97.8   | 3.995    | 78.35          | 65.44         |
> | + VLA-Cache                         | **98.3**   | **3.043**    | **61.12**          | **81.67**         |
> | + Token Merging                     | 83.6   | 3.187    | 89.51          | 63.53         |
> | + Token Pruning                     | 93.4   | 3.128    | 63.74          | 82.43         |
>
> As VLA models are highly sensitive to subtle visual cues, both *token merging* and *pruning* tend to disrupt spatial consistency, an essential requirement for robotic manipulation. Our results in Table 4 of the main paper further confirm that applying these methods leads to a noticeable drop in success rate and inconsistent efficiency improvements.
>
> In contrast, our **reuse-based caching strategy** is explicitly designed to preserve spatial integrity by avoiding token alteration and instead reusing computation only for static regions. While technically compatible, combining VLA-Cache with merging or pruning introduces instability and harms performance. Therefore, we recommend using our reuse strategy independently to maintain both accuracy and efficiency.
>
> **Reference:**
>
> [3] Zhang, Y., et al. (2024). Sparsevlm: Visual token sparsification for efficient vision-language model inference. ICML 2025
> [4] Chen, L.,  et al. (2024). An image is worth 1/2 tokens after layer 2: Plug-and-play inference acceleration for large vision-language models. ECCV 2024
>
> **W4. Contribution is local rather than fundamental**
>
> **R5:**
> We respectfully clarify that our contribution extends beyond engineering optimization. Our work addresses a critical and underexplored challenge in the field: the inference inefficiency of billion-scale VLA models in real-time robotics, which currently limits their practical deployment.
>
> Rather than implementing a software-level tweak, VLA-Cache is a modular, architecture-agnostic algorithm carefully designed to exploit intrinsic structural properties of VLA models. It integrates three steps including static token detection, task-relevance filtering, and layer-adaptive reuse. Importantly, our method is plug-and-play: it requires no retraining or fine-tuning, and generalizes across diverse VLA architectures.
>
> Beyond practical acceleration, VLA-Cache also offers insight into the learning and information flow of VLA models. It reveals that token-level redundancy and temporal locality are prominent, exploitable features. Selective reuse of stable representations preserves performance while substantially reducing computational cost. These findings open new directions for building efficient and deployable VLA systems.
>
> ---
>
> **We sincerely appreciate your valuable suggestions and hope our rebuttal can address your concerns. We are happy to answer any remaining questions during the discussion phase.**

---

> ### Author Response · Authors · 2025-08-06
>
> We sincerely thank you for your efforts in reviewing our paper. We have provided corresponding responses and results, which we believe have covered your concerns. We hope to further discuss with you whether your concerns have been addressed or not. Please let us know if you still have any unclear parts of our work.
>
> Best Regards,
>
> Authors of Submission 5408

---

> > ### Comment · Reviewer_bXk4 · 2025-08-07
> >
> > Thanks to the authors for their clarification, and apologies for the delayed response.
> > As the authors pointed out, I had overlooked the real robot experiments, and the rebuttal has effectively addressed my concerns. Considering the novelty, experimental results, and the additional explanations provided, I believe this work meets the quality standards with a score of 4. Therefore, I will maintain my score as weak accept.

---

### Official Review · Reviewer_4fFX · 2025-07-03

**Clarity:** 3
**Significance:** 3
**Originality:** 3
**Rating:** 4
**Confidence:** 3

**Summary:**

This paper considers a lazy strategy for computing K-V values in transformers, specifically for the usage of robotic VLA. The proposed algorithm finds the static tokens and filters those task-relevant. Experimental results show that the VLA-Cache technique accelerate existing VLA algorithms. Overall, I think the motivation of the paper is good.

**Questions:**

Please see my comments above.

**Ethical Concerns:**

["NO or VERY MINOR ethics concerns only"]

**Limitations:**

yes

**Paper Formatting Concerns:**

no such issues

**Quality:**

3

**Strengths And Weaknesses:**

1. I didn't find the opensourcing code of this paper. As is known to all, the code implementation matters a lot when computational efficiency is concerned. Will the authors opensource the code?

2. The proposed algorithm may not work if the robot sensing data is not 2D images, nor if the sensor is mounted on the robot rather than a third-view perspective. If the sensing data is point cloud, then the proposed algorithm is also inapplicable, because the data is unordered.

3. I think the proposed algorithm should also work on other kind of VLA modules, such as DP and RDT. Please consider add them.

---

> ### Author Rebuttal · Authors · 2025-07-31
>
> We sincerely thank you for your effort in reviewing our paper. Your insightful and helpful comments helped us for refining our paper better. Below, we address your comments and questions in detail.
>
> ---
>
> **Q1. Code release**
>
> **R1:** Thank you for raising this important point. To this end, we have included the full implementation of VLA-Cache in the **supplementary material** for verification.
>
> We are fully committed to open-sourcing the complete codebase upon paper acceptance, including scripts for reproducing all benchmarks and real robot deployment. We hope this will support further research and practical deployment.
>
> ---
>
> **Q2.1. Applicability to Point Cloud or Non-2D Inputs**
>
> **R2.1:**
> Thanks for this insightful question. Our current work focuses on vision-language-action (VLA) models that process 2D image inputs, which remain the dominant form across recent models (e.g., OpenVLA, OpenVLA-OFT, $\pi_0$) and large-scale datasets like Open X-Embodiment.
>
> We acknowledge that the proposed VLA-Cache is designed and evaluated in this 2D visual context. However, the core idea of **selective token reuse based on temporal visual redundancy is not inherently tied to 2D images**. For multi-modal VLA models (e.g., PointVLA [1]) that integrate point cloud data, VLA-Cache can still operate on the image branch. Moreover, similar principles such as identifying static or low-saliency subsets of unordered point cloud tokens, can also be extended to non-2D domains with appropriate adaptation. While evaluating this on point cloud inputs is beyond the current scope, we believe our method offers general insights into computation-efficient inference and is compatible with broader extensions in future work.
>
> **Reference:**
>
> [1] Li, C., et al. (2025). Pointvla: Injecting the 3d world into vision-language-action models. arXiv preprint arXiv:2503.07511.
>
> ---
> **Q2.2. Applicability to Robot-Mounted Cameras (e.g., Wrist Cameras)**
>
> **R2.2:** We validated VLA-Cache on OpenVLA-OFT, which includes inputs from both third-person and wrist-mounted (egocentric) cameras. As shown in Figure 4c/d and Table 2, the method maintains robustness under dynamic viewpoints and is able to reuse a significant portion of tokens even with egocentric inputs.
>
> ---
> **Q3. Applicability to Other VLA Architectures (e.g., DP or RDT)**
>
> **R3:**
> Thank you for this valuable suggestion. VLA-Cache is designed to accelerate the language decoder stage, and is therefore not directly applicable to VLA models without a VLM backbone, such as standalone DP-based architectures.
> However, for **hybrid architectures**, where a VLM encoder/decoder is combined with a diffusion policy head (e.g., CogACT [2]), VLA-Cache remains fully applicable. In fact, we have already validated our method on CogACT, achieving both efficiency and performance gains (please see Table 3).
>
> Beyond this, transformer-based models such as RDT or DiT also exhibit inter-frame redundancy, and our token-reuse strategy can be adapted to these settings as well. We consider this an exciting direction for future generalization.
>
> **Reference:**
>
> [2] Li, Q., et al. (2024). *Cogact: A foundational vision-language-action model for synergizing cognition and action in robotic manipulation*. arXiv:2411.19650
>
> ---
>
> **We sincerely appreciate your valuable suggestions and hope our rebuttal can address your concerns. We are happy to answer any remaining questions during the discussion phase.**

---

> ### Author Response · Authors · 2025-08-06
>
> We sincerely thank you for your efforts in reviewing our paper. We have provided corresponding responses and results, which we believe have covered your concerns. We hope to further discuss with you whether your concerns have been addressed or not. Please let us know if you still have any unclear parts of our work.
>
> Best Regards,
>
> Authors of Submission 5408

---

> > ### Author Response · Authors · 2025-08-09
> >
> > Thank you again for your valuable feedback and for engaging with our work. As the discussion deadline is approaching, we would like to check whether our previous responses have addressed your concerns. Please feel free to let us know if there are still any points that remain unclear, and we would be happy to clarify them.
> >
> > Best regards,
> >
> > Authors of Submission 5408

---

> ### Comment · Area_Chair_MUNr · 2025-08-09
>
> Thank you for your efforts of reviewing! Please remember to read the rebuttal as soon as possible, and discuss with the authors for questions not resolved. According to the guideline this year, reviewers need to participate in discussions with authors before submitting “Mandatory Acknowledgement”.

---

### Comment · Area_Chair_MUNr · 2025-08-04

Thank all the reviewers for your efforts of reviewing! Please remember to read the rebuttal, and discuss with the authors for questions not resolved.

---

### Note · Authors · 2025-08-12

Dear Area Chair and Reviewers,

Thank you for your dedicated efforts and the highly constructive engagement throughout the review process.

Below, we summarize the key strengths of our work, as highlighted by the reviewers, and outline how we have addressed their initial concerns.

**(1) Key Strengths**

We are encouraged that the reviewers recognized the core value of our contributions across several dimensions:

- Motivation & Clarity:
  - `4fFX`: *"I think the motivation of the paper is good"*
  - `zsdX`: *"the paper is clear and easy to follow"*
  - `B8wU`: *"the motivation for the proposed method clearly and understandably."*

- Method & Novelty:
  - `zsdX`: *"The idea of adaptively reusing the KV cache is interesting and seems to be novel"*
  - `B8wU`: *"This paper introduces a novel training-free inference acceleration method"*
  - `bXk4`: *"Clear innovation", "Well-designed method", "Clever integration of visual staticness and semantic importance"*

- Experiments & Results:
  - `B8wU`: *"The paper establishes and conducts extensive experiments"*
  - `bXk4`: *"Comprehensive evaluation metrics", "Strong baselines and comparisons"*

- Presentation:
  - `B8wU`: *"The paper is well-organized and features a clear and comprehensible writing style"*
  - `zsdX`: *"the paper is clear and easy to follow"*

**(2) Post Rebuttal**

- `4fFX`  initially provided a score of 4. Although no direct discussion took place, we addressed the issues of non-2D inputs, wrist camera and other VLAs in our rebuttal, and we appreciate the reviewer’s time and consideration.

- `bxk4` confirmed that they had overlooked the real-world robot experiments, and that our rebuttal effectively addressed their concerns. They now believe that “**this work meets the quality standards with a score of 4.**” We appreciate the reviewer’s time and consideration. *This update does not yet appear in the system, and we kindly hope the reviewer can update the rating.*

- `B8wU` stated that "**your responses have satisfactorily addressed most of my concerns, and as a result, I have raised my score to a 4**". We appreciate the reviewer's time and consideration. *This update seems not yet reflect in the system, and we kindly hope the reviewer can update the rating in the system.*

- `zsdX` acknowledged our rebuttal and stated, "**Most of my previous questions are clarified. I would like to maintain the positive score.**"

Thank you once again for your time and your valuable role in improving our work.

---

### Decision · Program_Chairs · 2025-09-17

**Decision:**

Accept (poster)

**Comment:**

This paper proposes VLA-Cache, a training-free inference acceleration method for Vision-Language-Action (VLA) models. It reduces redundant computation by reusing static visual tokens across frames, while selectively recomputing task-relevant tokens and applying layer-adaptive reuse. The paper is clearly written and well motivated. Experiments on LIBERO, SIMPLER, and a real robot report up to 1.7× CUDA speedup and 15% higher control frequency with minimal loss in success rate.

**Strengths**:
- Clear writing and strong motivation.
- Novel training-free acceleration method with three complementary components.
- Extensive experiments across simulation and real-world platforms.

**Weaknesses**:
- Real-world reporting needs clarification. The rebuttal states 20 trials for PickPot, but the PlaceCube success rate (e.g., 83.3%) implies a different number of trials。

**Rebuttal**:
The authors clarified applicability to wrist-camera settings (reuse ratio comparable to third-person), added ablations using attention scores as task-relevance proxies, and clarified comparisons with diffusion-based VLA methods.

**Suggestions for Improvement**:
- Include additional ablations and comparisons (e.g., attention-based proxies, diffusion-based VLAs) in the final version or appendix.
- Clarify real-world experimental details (trial counts, protocols, and success criteria).
- Correct Figure 4’s top-left label (should be LIBERO, not SIMPLER).